# Characterization of Al Incorporation into HfO_2_ Dielectric by Atomic Layer Deposition

**DOI:** 10.3390/mi10060361

**Published:** 2019-05-30

**Authors:** Md. Mamunur Rahman, Jun-Gyu Kim, Dae-Hyun Kim, Tae-Woo Kim

**Affiliations:** 1School of Electrical Engineering, University of Ulsan, Ulsan 44610, Korea; rahman.mamun37@gmail.com; 2School of Electronics Engineering, Kyungpook National University, Daegu 702-701, Korea; mhg095321@gmail.com

**Keywords:** border trap, HfAlO, high-*k*, interface trap

## Abstract

This study presents the characteristics of HfAlO films for a series of Al incorporation ratios into a HfO_2_ dielectric by atomic layer deposition on a Si substrate. A small amount of Al doping into the HfO_2_ film can stabilize the tetragonal phase of the HfO_2_, which helps to achieve a higher dielectric constant (*k*) and lower leakage current density, as well as a higher breakdown voltage than HfO_2_ film on its own. Moreover, assimilation of Al_2_O_3_ into HfO_2_ can reduce the hysteresis width and frequency dispersion. These are indications of border trap reduction, which was also verified by the border trap extraction mechanism. X-ray photoelectron spectroscopy (XPS) analysis also verified the HfAlO microstructural properties for various Al compositions. In addition, higher amounts of Al_2_O_3_ in HfAlO resulted in better interface and dielectric behavior through trap minimization, although the equivalent-oxide-thickness (EOT) values show the opposite trend.

## 1. Introduction

The traditional dielectric material SiO_2_ has inherent chemical and thermal stability on Si wafers, in addition to a large band gap and elevated barrier height between the dielectric and Si [1,2,3]. However, SiO_2_ thickness has reached an optimum as an insulating material due to miniaturization of metal-oxide-semiconductor field-effect-transistors (MOSFETs). Thus, low-power application devices mainly suffer from higher leakage current due to direct tunneling through the SiO_2_ film [4]. However, a thicker oxide layer with a high dielectric constant (*k*) can decrease the leakage current and the equivalent-oxide-thickness (EOT), which are related to the speed of the transistor.

Among the different deposition techniques such as physical-vapor-deposition (PVD), chemical-vapor-deposition (CVD), and atomic-layer-deposition (ALD) for high-*k* oxides, ALD [5,6,7,8,9] is considered the most promising for realization of microelectronics and nanotechnology. ALD is a subclass of CVD and is unique in that the precursors and oxidants are not present in the deposition chamber at the same time. Rather, the precursor and oxidant are introduced into the chamber in a sequential and non-overlapping way. Between the precursor and oxidant injections, a purge gas flow is maintained to remove the remaining precursor and reactant species. The combination of a precursor/oxidant pulse and purge gas flow is known as a half-cycle. Film deposition occurs through a self-saturating half-cycle since the reactions terminate by themselves once all reactive elements on the surface are consumed [9]. Therefore, unlike CVD where the deposition occurs on a time basis, in ALD it occurs on a cyclic basis [5]. The thickness of the deposited film can be adjusted according to the number of deposition cycles. Thus, the self-saturating nature and cyclic deposition provides a conformal, controlled, uniform high-quality, dense, and pinhole-free thin film deposition with a thickness error of less than 1%; this film growth is independent of precursor and oxidant fluxes [7,10]. A detailed comparison of different deposition techniques, including ALD, is described in [9]. 

Among the reported high-*k* oxides, Al_2_O_3_ and HfO_2_ are considered the most promising replacements for SiO_2_ because of their ease of manufacture by ALD [11]. Al_2_O_3_ has a larger band gap (~8.8 eV) and band alignment similar to SiO_2_, good thermal stability, and reduced oxygen and ionic transport, but it suffers from a moderate dielectric constant (*k* = 6–9) [12]. On the other hand, HfO_2_ has a higher dielectric constant (*k* = 20–25) and moderate band gap (~5.68 eV), but it suffers from low crystallization temperature, poor thermal stability, and poor blocking of oxygen diffusion through the HfO_2_ [13,14]. It has been reported that incorporation of Al_2_O_3_ into HfO_2_ in the form of a bilayer (Al_2_O_3_/HfO_2_) can increase the thermal stability up to 900 °C; this value can increase further to 1000 °C when the material is in the form of a nanolaminate (HfAlO) [15,16]. Al incorporation also increases the dielectric constant and band gap while reducing hysteresis and electrical defects [13,17,18]. In addition, incorporation of Al in alloy form (HfAlO) provides better EOT scaling, lower interface state density (D_it_), and a diminished gate leakage current in comparison to the bilayer arrangement [19,20]. As a result, HfAlO has potential benefits, including simultaneous lower interface state density and EOT scaling.

Among the available HfAlO reports, most of the cases involve incorporation of Al into HfO_2_ by maintaining some discrete Al: Hf ratio. Therefore, it is difficult to consider the overall scenario based on Al composition. In addition to the interface state density (D_it_), the border trap density (N_bt_) of the dielectric is currently attracting attention. Border traps are those inside the dielectric, especially in high-*k* oxides, that cause dispersion in the accumulation region at the capacitance–voltage (C–V) behavior due to charge transport between the semiconductor and insulating material via tunneling [21,22,23]. These traps affect the on-state device performance by threshold voltage shifting and degradation of transconductance [23]. The interface states have very small time constants that cannot retort in the typical measurement frequency range (1 kHz to 1 MHz) at functional voltage; thus, the conventional interface state phenomenon is unable to explain these observations [24]. It has already been reported that HfO_2_ suffers more severely from these types of traps than does Al_2_O_3_, and that Al incorporation into HfO_2_ effectively reduces border traps [25,26]. In this paper, a change in the electrical behavior of HfAlO dielectric films was achieved by varying the Al insertion amount from 10–50%. In addition, quantification of the interface and border trap density was performed according to the Al: Hf ratio. 

## 2. Materials and Methods 

Single Al_2_O_3_, HfO_2_, and HfAlO alloy thin films were deposited by a thermal ALD system, at a deposition temperature of 250 °C, on n-type Si (100) substrate (SEHYOUNG Wafer Co. Ltd., Seoul, Korea) with resistivity of 1–10 Ω-cm using trimethylaluminum (TMA) and tetrakis (ethymethylamino) hafnium (TEMAH) as metal precursors of Al_2_O_3_ and HfO_2_, respectively, while H_2_O was used as an oxidant. N_2_ was used as both the carrier and purge gas. The Chemical Abstracts Service (CAS) number for TMA is 75-24-1, while that for TEMAH is 352535-01-4. Both precursors were claimed to have a 99.9999% purity as provided by UP Chemical Co., Ltd. (Pyeongtaek, Korea). The ALD system is a thermal ALD process (“Atomic Classic”, CN1, Hwaseong, Korea), having a maximum deposition temperature of 450 °C and an allocation system of 4 sets of precursor canisters. The TMA precursor canister was kept at room temperature, and the TEMAH precursor was at 60 °C. The carrier and purge gas flow rates were 300 sccm for Al_2_O_3_ deposition and 100 sccm for HfO_2_. The pulse times were 0.1 s and 2 s for Al_2_O_3_ and HfO_2_ precursors, respectively, while the purge time was 20 s in both cases. The oxidant (H_2_O) pulse times were 0.1 s and 0.2 s for Al_2_O_3_ and HfO_2_ cases, respectively. All the depositions were performed at a chamber pressure of 3.6 millitorr. Before deposition, the substrates were processed by standard pre-cleaning and diluted hydrofluoric acid (HF) (CAS number:7664-39-3, purity:49.2%, J.T. Baker-a division of Mallinckrodt Baker Inc.; Phillipsburg, NJ, USA) stripping for native oxide removal. They were then rinsed in deionized (DI) water and finally dried in a N_2_ environment for the prevention of water mask formation on the surface. A total of seven HfAlO alloy oxide samples were grown following the supercycle concept for precise control of composition, where all deposition started with Al_2_O_3_ to improve the interface quality [13]. The deposition cycles consisted of a pre-nitrogen purge as a first step, m cycles of Al_2_O_3_ deposition as a second step, n cycles of HfO_2_ deposition as step 3, and finally a post-nitrogen purge as step 4. Steps 2 and 3 together are called supercycle and were repeated to form (m. Al_2_O_3_ + n. HfO_2_) x, where m is the number of Al_2_O_3_ cycles, n is the number of HfO_2_ cycles, and x is the quantity of supercycle. The number of supercycle was adjusted according to the calculated growth rate-per-cycle (GPC) of Al_2_O_3_ and HfO_2_ obtained from 50 individual cycle depositions to maintain a similar level of thickness. The thickness of the ALD deposited films was measured by ellipsometry at an incident angle of 70°. Design of the supercycle and the thickness summary are listed in Table 1. An X-ray photoelectron spectroscopy (XPS) system (ThermoFisher, Waltham, MA, USA, model: K-alpha) equipped with monochromatic Al Kα with a photon energy of 1.4866 keV was used for examination of microstructural properties. The surface morphology investigation was performed using an atomic force microscopy (AFM) system (WITec, Ulm, Germany, model: alpha300S). For MOSCAP device formation, a 1500 Å thick aluminum (Al) metal layer was deposited by e-beam evaporation (Temescal, Zeus Co, Ltd.; Yongin, Korea, model: FC-2000) on the dielectric for a front electrode with an area of 2.5 × 10^−4^ cm^2^ using a shadow mask. The same type of metal layer was also deposited as a backside contact. Electrical characterizations, including capacitance-voltage and current-voltage measurements, were performed using a Keithley 4200A-SCS parameter analyzer (Tektronix, Inc., Beaverton, OR, USA) at a sufficient gate voltage.

## 3. Results and Discussion

Figure 1a shows the capacitance–voltage (C–V) characteristics at high frequency (1 MHz) for the HfAlO MOS capacitors with single Al_2_O_3_ and HfO_2_ capacitors. The solid lines represent the capacitance values of Al_2_O_3_ and HfO_2_ MOSCAPs, while the dotted lines signify the capacitance of HfAlO alloy cases. The figure shows that the single Al_2_O_3_ (sample A) MOSCAP had the lowest capacitance, whereas sample F, which had an Al: Hf ratio of 1:9, showed the maximum capacitance. However, without these two extreme points, the other C–V curves cannot explain the full scenario due to the different thicknesses, as mentioned in Table 1. For all the samples, thickness measurement was done at five different points on a wafer and the thickness value represented here is the average thickness of the five points with the standard deviation. According to the thickness values listed in Table 1, the single HfO_2_ layer (sample I) had the lowest thickness, while the HfAlO alloy with an Al: Hf = 3:3 ratio (sample H) had the highest. These data indicate that the thickness clearly depends on Al content. The values for the other samples are within these ranges. This thickness difference is mainly due to the different GPCs of Al_2_O_3_ and HfO_2_ as well as the different numbers of supercycle, which depend on the Al: Hf ratio. Since the supercycle consists of both Al_2_O_3_ and HfO_2_ deposition, cycles had to be completed (rather than comparing precise thicknesses) in some cases to avoid the fractional case. The multifrequency (10 kHz–1 MHz) capacitance–voltage responses are plotted in Figure 1b. From the figure, it is evident that the Al inclusion has minimized the C–V dispersion in accumulation which represents the border trap reduction. Also, in highly HfO_2_ rich samples (samples D, E, F, I) there is some hump in the weak inversion region, which depicts a high number of interface state traps [27]. In addition, it is also apparent from the figure that the flat band voltage shifted in the positive direction after incorporation of Al due to the addition of negatively charged dipoles [28].

(1)$$EOT_{total}=A\frac{\varepsilon_{0}\varepsilon_{SiO_{2}}}{C_{ox}}$$

(2)$$EOT_{high-k}=EOT_{total}-T_{SiO_{2}}$$

(3)$$\varepsilon_{high-k}=\frac{T_{high-k}}{EOT_{high-k}}\varepsilon_{SiO_{2}}$$

The impact of Al incorporation can be better understood by extracting the dielectric constant (*k*) and equivalent-oxide-thickness (EOT) of all the samples. The dielectric constant (*k*) and EOT are calculated by the following equations as described in [29]:

Here, A is the area of the electrode, C_ox_ is the accumulation capacitance measured at 1 MHz, T_high-k_ and Tsio_2_ are the thickness of high-*k* dielectric layer and SiO_2_ layer, respectively, and ε_high-k_ and ε_sio2_ are the permittivity of the high-*k* layer and SiO_2_ layer. The thickness of the as-grown SiO_2_ layer during ALD deposition was measured at ~0.8 nm using ellipsometry, and its permittivity was considered to be 3.9. Figure 1c illustrates the summary of the dielectric constant (*k*) and EOT of all splits. Pure Al_2_O_3_ and HfO_2_ had dielectric constants of 5.78 and 16.64, respectively, which are a little lower than the reported ones. This may be attributed to formation of some interfacial layer (IL). The dielectric constant (*k*) improved with higher Hf fraction in the alloy films. Sample E (Al: Hf = 1:4) showed a slightly higher *k* value than HfO_2_, which was 17.42, and sample F (Al: Hf = 1:9) showed the highest value (21.23). The increase in the dielectric constant of HfAlO films was attributed to the change in microstructure caused by the phase transition of the deposited film. It has been reported that the smaller radius Al atom can diffuse into HfO_2_ film, which can change the HfO_2_ phase from monoclinic to tetragonal, which has a higher dielectric constant [*k*] [13,30]. Interestingly, sample B (Al: Hf = 1:1) showed the lowest permittivity among the HfAlO films, although the apparent formulas of sample B (Al: Hf = 1:1) and sample H (Al: Hf = 3:3) are the same. This may be attributed to the change in film stoichiometry and in Al and Hf mole fractions according to supercycle design. The EOT values followed a trend similar to those of permittivity.

Figure 2a illustrates a comparison of the measured C–V hysteresis of Al-incorporated HfO_2_ film-based MOSCAPs along with Al_2_O_3_ and HfO_2_ ones. The C–V hysteresis was measured by performing the sweep from inversion to accumulation and (without any delay) again sweeping back to inversion at a frequency of 1 MHz to exclude the response of interface traps. It is obvious from the figure that Al incorporation reduces the hysteresis of HfO_2_ due to the charges trapped at the oxide/semiconductor interface, as well as inside the dielectric. This reduction can be explained as follows, Al_2_O_3_ exhibits micromolecular properties, while TMA has high reactivity, which allows the pinholes of HfO_2_ to be filled with Al ions and promotes formation of a more condensed film during ALD deposition [31]. Interestingly, sample H (3:3) showed slightly higher hysteresis than expected. The surface charge density as illustrated in Figure 2b was quantified according to the following equation:
(4)$$Q_{trapped}=\frac{C_{ox}\times \Delta V}{q}$$
where Q_trapped_ denotes the trapped charge density per cm^2^, C_ox_ represents the oxide capacitance in F/cm^2^, ΔV is the voltage shift as measured at V, and q is the elementary electron charge in C. This surface charge includes the charges located near or at the dielectric/semiconductor interface, which are differentiated as fixed charge, ionized traps, and mobile ions. These fixed charges cause a C–V curve shift along the voltage axis in the parallel direction, and the two-dimensional distribution of these traps contributes to the overall capacitance. Since the high frequency hysteresis measurement excludes the small signal voltage information, it is represented in cm^−2^ rather than cm^−2^ eV^−1^. However, it is clear from the figure that the trapped charge density follows almost the same pattern as Figure 2a, since it is proportional to the hysteresis width even though it is affected by different C_ox_ values. The single Al_2_O_3_ layer showed the smallest number of traps (1.81 × 10^11^/cm^2^), and HfO_2_ had the highest (1.72 × 10^12^/cm^2^), while HfAlO samples were within these two boundary points. Sample B (Al: Hf = 1:1) alloy showed the lowest trap density among the HfAlO alloys, while sample F (Al: Hf = 1:9) showed the largest. Sample H (3:3) also showed a higher value, as previously mentioned.

Figure 3 shows the X-ray photoelectron spectroscopy (XPS) analysis for HfAlO films to understand the bonding structure and film stoichiometry. Among all the HfAlO splits, we chose two cases based on Al_2_O_3_ concentration according to the deposition cycle ratio of Al_2_O_3_ and HfO_2_. Figure 3a–c shows the Hf-4f, Al-2p, and O-1s core level spectra for sample G, which has an Al: Hf = 2:6 ratio and is supposed to have a higher Al_2_O_3_ concentration. Figure 3a for the Hf-4f spectra shows that the two peaks of 4f_7/2_ and 4f_5/2_, which are located at 17.52 eV and 18.9 eV, respectively, correspond to Hf bonding to oxygen, as reported in [32]. The higher binding energy movement of Hf-4f peaks compared with the reported pure HfO_2_ cases was attributed to Al incorporation, which changes the bonding characteristics due to formation of Hf-Al-O bonds [33,34,35]. This shift is because the electron density to the Al-O bond in the HfAlO alloy increased due to the difference in electron negativity of Hf, Al, and O [36]. The Al-2p core level spectra in Figure 3b were also deconvoluted into two peaks at 74.42 eV and 74.88 eV, which correspond to Hf-Al-O and Al-O bonds, respectively [37]. Similarly, after deconvolution of O-1s spectra, two peaks appear at 530.78 eV and 531.68 eV. The lower peak indicates the Hf-Al-O bond based on an amalgamation of Al_2_O_3_ and HfO_2_ layers, and its position is between those of pure HfO_2_ and Al_2_O_3_ [17]. The higher peak corresponds to C-O bonding due to surface contamination by hydrocarbons [32]. However, the same core level XPS spectra for sample F having Al: Hf = 1:9 are shown in Figure 3d–f. The peaks follow the same configuration as in previous cases, although the binding energy shifted to slightly lower values in almost all cases. This is due to a reduction in Al concentration as dictated by deposition cycle ratio, where the binding energies experience a similar shift due to a decrease of Al_2_O_3_ in the structure, as reported in [38]. Table 2 summarizes the bond binding energies according to peak position in the core level spectra. 

Figure 4 illustrates the 3D AFM images for surface morphology analysis in tapping mode on the HfAlO samples, with a scan area of 1 μm × 1 μm and a resolution of 256 points × 256 lines. For better understanding, both Al_2_O_3_ and HfO_2_ films were also analyzed. The root-mean-square (RMS) and the average roughness (R_a_) of the surface in the figures indicate the roughness measurement of the films. It is observed that the surface of the Al_2_O_3_ is quite flat and uniform compared to the HfO_2_ surface. The RMS value for Al_2_O_3_ is 55.77 pm while for HfO_2_, it is 75.55 pm. The HfO_2_ films with Al incorporated show a smoother surface than the HfO_2_ film without Al. Among the HfAlO films, sample H (3:3) has the lowest RMS roughness (57.18 pm) while sample F (1:9) has the highest (70.69 pm). The average roughness (R_a_) follows the same trend as RMS roughness. The decrease of surface roughness in the HfAlO film compared to the HfO_2_ film indicates a structural change.

Figure 5 shows the interface trap density and border trap characterization along with an illustration of frequency dispersion. The conductance method was used to calculate the interface traps by measuring the parallel conductance (G_p/ωmax_) with series resistance correction [39]. Then, the interface trap density (D_it_) was calculated using the following equation:(5)$$D_{it}=2.5\frac{{\left(G_{p}/\omega \right)}_{\max}}{Aq}$$
Here, A is the electrode area, while q is the elemental electron charge. From Figure 5a, it is evident that Al incorporation helps to passivate the dangling bonds as the HfAlO films showed fewer defects than the pure HfO_2_ film. Also, the increased Al amount in the HfAlO film provides a better interface as the single Al_2_O_3_ showed the lowest D_it_ value. Moreover, characterization of border traps (N_bt_) mainly highlights the accumulation frequency dispersion rather than hysteresis width. It has been reported that border traps are only probed by accumulation frequency dispersion around a solitary energy level at the maximum accumulation bias voltage based on the assumption of spatial distribution of traps inside the oxide; in contrast, border traps with energy levels below the fermi energy at the maximum bias voltage are filled during C–V hysteresis measurements, and these traps are at such distances that they do not re-emit the captured charges during the reverse sweep hysteresis analysis [25]. The frequency dispersions shown in the inset of Figure 5b are calculated according to the following equation:(6)$$\%Disp=\frac{\left(C_{\mathrm{low}}-C_{\mathrm{high}}\right)}{C_{\mathrm{high}}*\log(\frac{\omega_{\mathrm{high}}}{\omega_{\mathrm{low}}})}$$
Here, C_low_ is the capacitance value at ω_low_, which is 10 kHz, while C_high_ is for a ω_high_ value of 1 MHz. The border traps were extracted using the distributive border trap model suggested by Yuan et al. [40]. Specifically, we determined the best fit among the measured capacitance and the model capacitance value obtained by numerically solving the following equation: (7)$$\frac{dy}{dx}=-\frac{Y^{2}}{j\omega \varepsilon_{ox}}+\frac{q^{2}N_{bt}\ln(1+j\omega \tau )}{\tau }$$

The above equation has a boundary condition $Y=j\omega C_{s}$ at x = 0, where Y is the total admittance at any distance x from the oxide–semiconductor interface, ω is the angular ac frequency, and C_s_ is the semiconductor capacitance at any surface potential ψ_s_. N_bt_ is the volume concentration of oxide border traps at a distance x from the oxide–semiconductor interface at any energy level, and τ is the average time constant for electron capturing. The effective electron masses of the Al_2_O_3_, HfO_2_, and HfAlO films were 0.23 m_0_, 0.22 m_0_, and 0.18 m_0_, respectively, where m_0_ is the electron mass at rest [25,41]. The semiconductor capacitance C_s_ was measured at the extraction voltage of border traps by the one-dimensional Poisson-Schrodinger solver simulation tool (Nextnano) [42]. From the figure, it is evident that incorporation of Al into HfO_2_ reduces the traps inside the dielectric by filling it with TMA atoms, as discussed earlier, which is depicted in the frequency dispersion and border trap comparison curves, where both were calculated at 2V. Here also, the HfAlO alloy with the same apparent composition showed some differences, while single Al_2_O_3_ and HfO_2_ occupy the extreme points of the border traps at 2.35 × 10^19^ cm^−3^ ev^−1^ and 1.35 × 10^20^ cm^−3^ ev^−1^, respectively. All HfAlO alloys are within the boundaries set by the pure components.

Figure 6 shows the leakage current densities and breakdown voltage characteristics of HfAlO alloy films with Al_2_O_3_ and HfO_2_ films after applying a positive bias voltage. The HfO_2_ film showed a two-order-higher leakage current and the lowest breakdown voltage of 4.8 MV/cm (located around 2.43 V). Single Al_2_O_3_ showed a leakage current in the 10^−11^ A/cm^2^ range and a breakdown voltage around 8.2 MV/cm. It is evident that Al incorporation into HfO_2_ reduced the leakage current and increased the breakdown voltage by providing a defect-free dielectric (as discussed earlier), greater dangling bond passivation, and increased band gap. Here, sample B [1,1] and sample H [3:3] showed higher breakdown voltages as they have higher thicknesses and greater Al content based on deposition cycle design.

## 4. Conclusions

The behavior of the HfAlO alloy with respect to the Al: Hf ratio was verified, along with that of the single Al_2_O_3_ and HfO_2_ stacks. Lower Al incorporation increased the dielectric constant of HfO_2_ film though phase transition, while relative permittivity decreased with an increasing Al_2_O_3_ amount. Furthermore, leakage current decreased with breakdown voltage augmentation by Al_2_O_3_ assimilation since it provides better trap minimization and a bandgap increase for the pure HfO_2_ film. Both the border trap and interface traps were better minimized with an increased amount of Al_2_O_3_ in the HfAlO alloy, and a similar trend was observed in hysteresis and frequency dispersion analyses. In almost all cases, the HfAlO alloy showed behavior within the range set by single Al_2_O_3_ and HfO_2_ films. 

## Figures and Tables

**Figure 1 micromachines-10-00361-f001:**
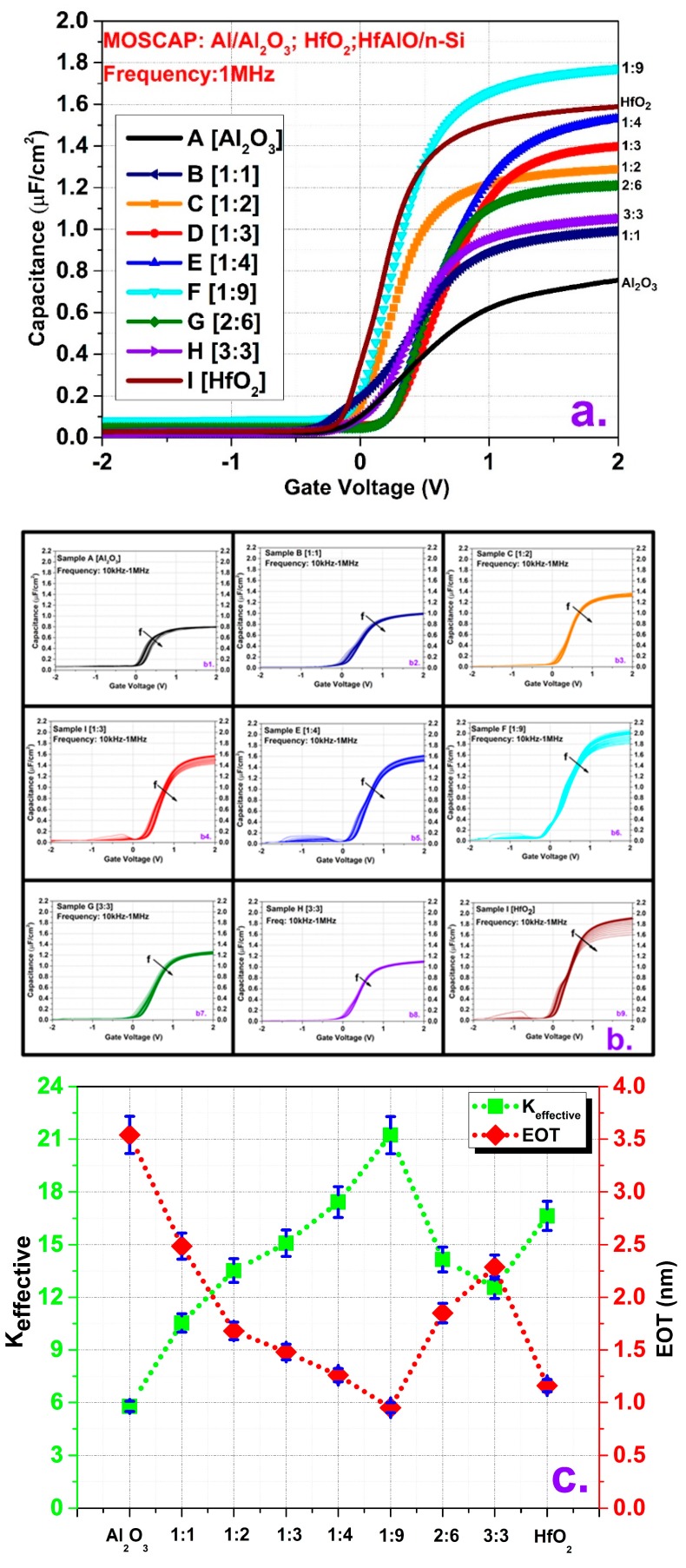
(**a**) Capacitance–voltage (C–V) behavior of the HfAlO MOS capacitors with single Al_2_O_3_ and HfO_2_ ones at 1 MHz frequency. (**b**) Multifrequency (10 kHz–1 MHz) capacitance–voltage (C–V) response of all deposition cases. (**c**) Extracted dielectric constant (*k*) and equivalent-oxide-thickness (EOT) values for all reported splits.

**Figure 2 micromachines-10-00361-f002:**
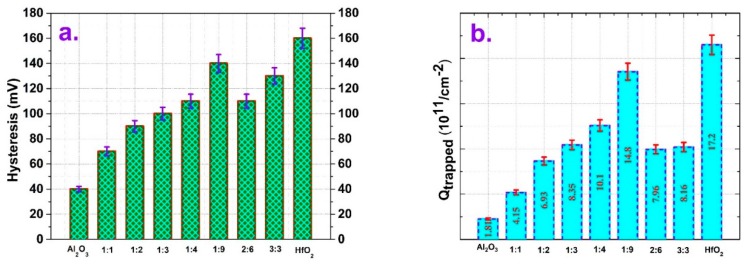
(**a**) Comparison of measured C-V hysteresis of Al/HfAlO/Si MOSCAPs with varying Al content (0–100%) at 1 MHz frequency. (**b**) Surface trapped charge density (from C–V hysteresis) of Al/HfAlO/Si MOSCAPs according to Al_2_O_3_ and HfO_2_ intermixing ratio.

**Figure 3 micromachines-10-00361-f003:**
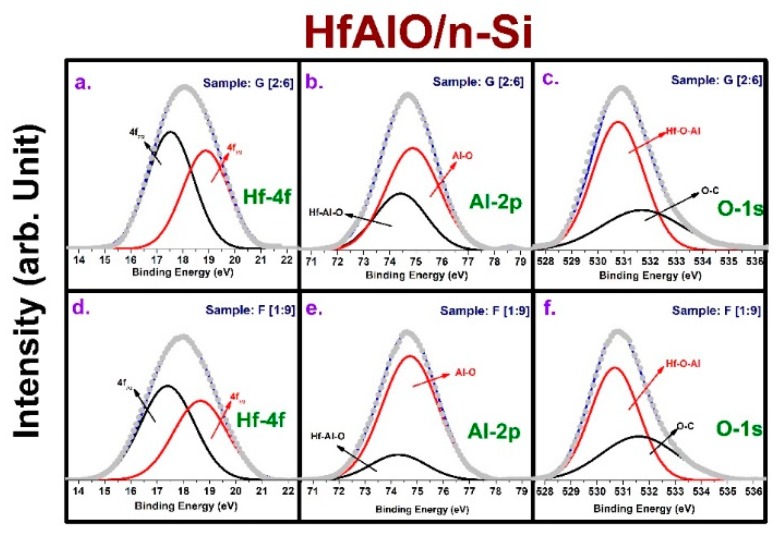
Hf-4f, Al-2p, and O-1s XPS core level spectra: (**a**–**c**) sample G (Al: Hf = 2:6) and (**d**–**f**) sample E (Al: Hf = 1:9).

**Figure 4 micromachines-10-00361-f004:**
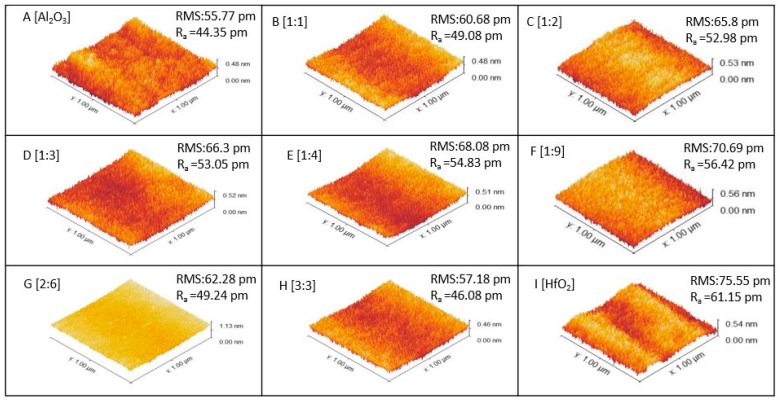
3D AFM images for the HfAlO samples, along with Al_2_O_3_ and HfO_2_ stacks. AL incorporation into HfO_2_ provides a smoother surface.

**Figure 5 micromachines-10-00361-f005:**
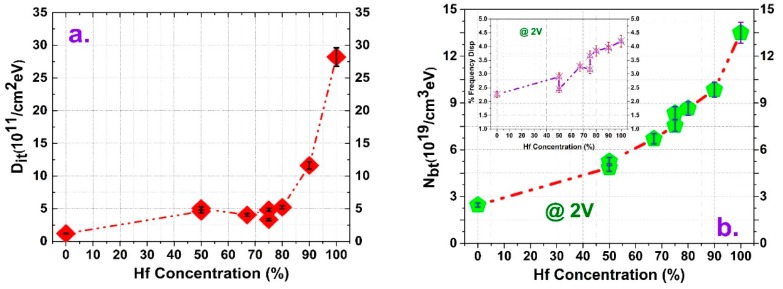
Trap characterization of HfAlO films with different Hf concentrations assumed by atomic-layer-deposition (ALD) cycle ratio: (**a**) interface trap density (D_it_) and (**b**) border tap density (N_bt_). Inset: frequency dispersion of accumulation capacitance at an applied bias of 2 V.

**Figure 6 micromachines-10-00361-f006:**
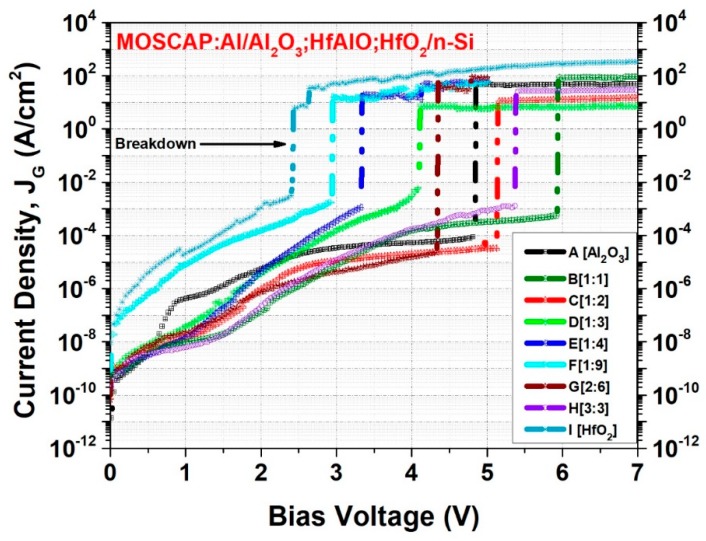
Leakage current density and breakdown voltage measurements under applied positive bias voltage for all the MOS capacitors.

**Table 1 micromachines-10-00361-t001:** Supercycle design summary for formation of HfAlO alloy along with deposited physical thickness.

Samples ID (m,n)	Al_2_O_3_ (m)	HfO_2_ (n)	Number of Super Cycles (x)	Thickness (nm)	Standard Deviation (nm)
A (1,0) [Al_2_O_3_]	1	0	50	5.867	0.125
B (1,1)	1	1	25	6.715	0.040
C (1,2)	1	2	17	5.825	0.063
D (1,3)	1	3	13	5.722	0.095
E (1,4)	1	4	10	5.626	0.101
F (1,9)	1	9	5	5.170	0.043
G (2,6)	2	6	7	6.710	0.022
H (3,3)	3	3	9	7.363	0.093
I (0,1) [HfO_2_]	0	1	50	5.100	0.089

**Table 2 micromachines-10-00361-t002:** The binding energies of different bonds according to peak position.

Samples ID	Al-O [Al-2p] (eV)	Hf-O [4f_7/2_; 4f_5/2_] (eV)	Hf-Al-O [Al-2p; O-1s] (eV)	C-O [Al-2p; O-1s] (eV)
G (2,6)	74.88	17.52; 18.9	74.42; 530.78	531.68
F (1,9)	74.72	17.4;18.65	74.3; 530.68	531.6

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
