# Peer review of "Characterization of Al Incorporation into HfO2 Dielectric by Atomic Layer Deposition"

_micromachines, 2019, doi:10.3390/mi10060361_

Round 1
Reviewer 1 Report
The manuscript describes the electrical behaviour HfAlO films with different stack ratios´. The paper is truly well written and organized and the results were well explained by a defined set of experiments. The paper is worthy of publication in Micromachines after careful addressing the following questions as minor issues.
Due to the increasing trend on flexible electronics and relevant fabrication challenges, there is an important need to focus on low-temperature processes including ALDs. It would be helpful to include the achievements if authors conduct a series of experiment on low-temperature deposition ALD (<100 °C)? If not, what would be the expected behaviours from authors' point of views?
Can authors provide more information about the roughness of deposited layers by conducting some AFM measurements and SEM imaging?
Due to the inevitable essence of native SiO2, it makes more sense to deposit a similar metal layer before oxide deposition to have a metal stack on both sides the oxide. Since even by a careful cleaning step, the native oxide layer will be formed immediately. Could authors comment on how they neglected the effect of formed native oxide?
It is also useful to have a verity of MOSs with different diameters to verify the quality of deposited oxides. Could authors comment why they chose the mentioned area without varying the size of MOS? Also, the capacitance measurement as a function of frequency would be required to qualify the deposited oxides. Do authors have some data on capacitance vs frequency of layers?
Authors need to explain why only two samples were studied by XPS and also it is helpful if they could summarise the XPS's achievement in a table.
Author Response
Reviewer 1
Reviewer wrote:
The manuscript describes the electrical behavior HfAlO films with different stack ratios. The paper is truly well written and organized and the results were well explained by a defined set of experiments. The paper is worthy of publication in Micromachines after careful addressing the following questions as minor issues.
Our response:
Dear the reviewer,
Thank you very much for carefully reviewing our manuscript and providing fruitful suggestions. We have taken all the comments into consideration, as below. We hope that the revision would be satisfactory to the reviewer and looking forward to hearing more comments.
Corresponding change in manuscript: No
Comment 1
Reviewer wrote:
Due to the increasing trend on flexible electronics and relevant fabrication challenges, there is an important need to focus on low-temperature processes including ALDs. It would be helpful to include the achievements if authors conduct a series of experiment on low-temperature deposition ALD (<100 °C)? If not, what would be the expected behaviours from authors' point of views?
Our response:
Thanks for your valuable comment. Yes, there are several reports of low temperature ALD deposition at below or around 100°C. Since our system is a thermal ALD system and typical temperature window in thermal system is 150°C-350°C. [H. B. Profijt, S. E. Potts, M. C. M. van de Sanden, and W. M. M. Kessels, Plasma-Assisted Atomic Layer Deposition: Basics, Opportunities, and Challenges, Journal of Vacuum Science & Technology A 29, 050801 (2011); doi: 10.1116/1.3609974]. Although our system is capable of deposition up to 450°C, we checked several deposition of Al2O3 and HfO2 within the typical temperature window. Among this range, we have found that at 250°C, a better dielectric constant and more uniform deposition with respect to others. Moreover, at 150°C, we have found that a higher GPC compered to nominal range which reflects the precursors and oxidant condensation on the film due to low thermal energy. Based on our previous experience, below 150 °C, we could get the oxide film for just dielectric materials for process purpose not for MOSCAP. We assumed that to deposit low temperature ALD deposition, lots of condition including precursor-type and reactant temperature should optimize for purpose.
Corresponding change in manuscript: No.
Comment 2
Reviewer wrote:
Can authors provide more information about the roughness of deposited layers by conducting some AFM measurements and SEM imaging?
Our response:
Thank you for pointing out this issue. We have performed the AFM measurements included the information in the manuscript. Since the information of roughness on our high-k, AFM images are enough to see on that information, so we didn’t include SEM images. We are sorry for our limitation.
Corresponding change in manuscript: Yes, we have included the AFM analysis and insert a new figure (Figure 4) and corresponding description.
New: Figure 4. 3D AFM images for the HfAlO samples along with Al2O3 and HfO2 stacks. Al incorporation into HfO2 provides a smoother surface.
Location of change:
Section: Result and Discussion
Page-9, line 232
Figure 4
Comment 3
Reviewer wrote:
Due to the inevitable essence of native SiO2, it makes more sense to deposit a similar metal layer before oxide deposition to have a metal stack on both sides the oxide. Since even by a careful cleaning step, the native oxide layer will be formed immediately. Could authors comment on how they neglected the effect of formed native oxide?
Our response:
Thank you for pointing this issue. If we deposit a similar metal layer before oxide deposition to have a metal stack on both side of the oxide, then the structure will be a Metal-Insulator-Metal (MIM) capacitor, which is quite different from our Metal-Oxide-Semiconductor (MOS) structure. In a MOS structure it is quite impossible to avoid the native oxide formation. But we have carried out diluted HF dipping for removing native oxide on Si surface. And then immediately loading the sample into ALD chamber. In addition, in our thickness calculation of we subtracted the native oxide thickness (achieving from without high-k samples) from the total thickness of the deposited samples. Also, in the calculations of dielectric constant and EOT we neglect the effect by using the Eqn. (1-3). Since interface traps and border traps are calculated by using the frequency dispersion data, we can’t fully neglect the effect of native oxide in these cases.
Corresponding change in manuscript: No.
Comment 4
Reviewer wrote:
It is also useful to have a verity of MOSs with different diameters to verify the quality of deposited oxides. Could authors comment why they chose the mentioned area without varying the size of MOS? Also, the capacitance measurement as a function of frequency would be required to qualify the deposited oxides. Do authors have some data on capacitance vs frequency of layers?
Our response:
Thank you for your comment. The answers are as follows
1. Yes, as you mentioned, it is useful MOSs with different areas. But we optimized the area with our shadow mask and decide stable condition on capacitance value. However, we have only one area for these sample which has been optimized before, so we couldn’t check for different area cases. We are really sorry for our limitation. There are also report available to verify using a single area. A. M. Mahajana, A. G. Khairnara, and B. J. Thibeault, Electrical Properties of MOS Capacitors Formed by PEALD Grown Al2O3 on Silicon, Semiconductors, Vol. 48, No. 4, 2014, pp. 497–500.
2. We are really sorry for our mistake for not-including this data. We have included the data of capacitance vs. frequency as Figure 1b.
Corresponding change in manuscript: Yes, we have included a new figure, named as Figure 1b. Previous Figure 1b is now Figure 1c.
New: Figure 1b: Multifrequency (10 kHz-1 MHz) capacitance-voltage (C-V) response of all deposition cases.
The multifrequency (10 kHz-1 MHz) capacitance-voltage responses are plotted in figure 1(b). From the figure, it is evident that, the Al inclusion has minimized the CV dispersion in accumulation which represents the border trap reduction. Also, in highly HfO2 rich samples (sample D, E, F, I) there is some hump in the weak inversion region, which depicts a high amount of interface state traps [28].
Location of change:
Section: Result and Discussion
Page-4, line 129 and Page-5, line 138
Figure 1(b)
Comment 5
Reviewer wrote:
Authors need to explain why only two samples were studied by XPS and also it is helpful if they could summarise the XPS's achievement in a table.
Our response:
Thanks for your comment. Our responses are below:
1. In this manuscript, we mainly focused on the electrical characterization of the deposited samples. We have performed the XPS analysis to be confirmed that HfAlO alloy has formed. Since all the samples would give same type of data set, so arbitrary we have chosen 2 samples (based on Al high and low concentration) to verify the HfAlO formation according to the Al incorporation amount.
2. We are sorry for not including the summary. We have summarized the XPS achievement in Table 2.
Corresponding change in manuscript: Yes, we have added Table 2.
Location of change:
Section: Result and Discussion
Page-9, line 227
Table 2
Reviewer 2 Report
The paper by Rahman et al. described the characterization of Al incorporation into HfO2 dielectric by Atomic layer Deposition on Si. This work is a very good contribution to the field and could be published in Micromachines after major revision as mentioned below:
1. What is the aim to precise in the title that ALD was performed on Si. I suggested that the title could be changed to: “Al incorporation into HfO2 dielectric by Atomic layer Deposition”
2. Some recent reviews in the field of ALD should be cited in the introduction: Biosensors and Bioelectronics, 2018, 122, 147-159 and Chemistry of Materials 2018, 30, 7368-7390
3. English should be improved
4. The advantages of using ALD for the manufacturing of high- k oxides in comparison to other methods should be discussed in the introduction.
5. CAs number, purities an providers of all the used chemicals should be added in the experimental section
6. The ALD equipment used to perform the deposition should be as well described
7. Error bar should be added to all values, tables and figures given in the manuscript. Is it relevant for instance to add 4 number after the decimal point for thicknesses in table 1?
8. Can the authors comment on the stability of the deposited films?
Author Response
Reviewer 2
Reviewer wrote:
The paper by Rahman et al. described the characterization of Al incorporation into HfO2 dielectric by Atomic layer Deposition on Si. This work is a very good contribution to the field and could be published in Micromachines after major revision as mentioned below:
Our response:
Dear the reviewer,
Thank you very much for carefully reviewing our manuscript and providing fruitful suggestions. We have taken all the comments into consideration, as below. We hope that the revision would be satisfactory to the reviewer and looking forward to hearing more comments.
Corresponding change in manuscript: No
Comment 1
Reviewer wrote:
What is the aim to precise in the title that ALD was performed on Si? I suggested that the title could be changed to: “Al incorporation into HfO2 dielectric by Atomic layer Deposition”
Our response:
Thanks for your valuable suggestion. Actually, our aim was to focus the substrate on which the deposition occurs. However, we have changed the title according to your suggestion.
Corresponding change in manuscript: The title has changed.
Previous: Characterization of Al incorporation into HfO2 dielectric by Atomic layer Deposition on Si
New: Characterization of Al incorporation into HfO2 dielectric by Atomic layer Deposition
Location of change:
Section: Title
Page-1 and line 2.
Comment 2
Reviewer wrote:
Some recent reviews in the field of ALD should be cited in the introduction: Biosensors and Bioelectronics, 2018, 122, 147-159 and Chemistry of Materials 2018, 30, 7368-7390
Our response:
Thank you for the comments. We are really sorry for not including recent references. We have added some recent references now.
Corresponding change in manuscript: Some references have been added in Introduction section.
New:
5. Clark, R. Emerging Applications for High K Materials in VLSI Technology. Materials 2014, 7, 2913–2944.
6. Croizier, G.; Martins, P.; Le Baillif, M.; Aubry, R.; Bansropun, S.; Fryziel, M.; Rolland, N.; Ziaei, A. Advantages of ALD over evaporation deposition for high-k materials integration in high power capacitive RF MEMS. In Proceedings of the 19th International Conference on Solid-State Sensors, Actuators and Microsystems (TRANSDUCERS); IEEE, 2017; pp. 1237–1240.
7. Mackus, A.J.M.; Schneider, J.R.; MacIsaac, C.; Baker, J.G.; Bent, S.F. Synthesis of Doped, Ternary, and Quaternary Materials by Atomic Layer Deposition: A Review. Chemistry of Materials 2019, 31, 1142–1183.
8. Knoops, H.C.M.; Faraz, T.; Arts, K.; Kessels, W.M.M. (Erwin) Status and prospects of plasma-assisted atomic layer deposition. Journal of Vacuum Science & Technology A 2019, 37, 030902.
9. Peter Ozaveshe Oviroh, Rokhsareh Akbarzadeh, Dongqing Pan, R.; Coetzee, A.M.; Jen, T.-C. New development of atomic layer deposition: processes, methods and applications. Science and Technology of Advanced Materials 2019.
Location of change:
Section: Introduction, References
Page-1, line 30 and References, line 334-345
Comment 3
Reviewer wrote:
English should be improved
Our response:
Thanks for your valuable suggestion. We have revised our manuscript with native speaker correction. And we submit the certificate of English correction for a reference with revised manuscript.
Corresponding change in manuscript: The English language has improved.
Comment 4
Reviewer wrote:
The advantages of using ALD for the manufacturing of high-k oxides in comparison to other methods should be discussed in the introduction.
Our response:
Thanks for your valuable suggestion. We are really sorry for not including this. We have included the description as your suggestion.
Corresponding change in manuscript: A paragraph has added.
New:
Among the different deposition techniques such as physical-vapor-deposition (PVD), chemical-vapor-deposition (CVD), and atomic-layer-deposition (ALD) for high-k oxides, ALD is [5–9] considered the most promising for realization of microelectronics and nanotechnology. ALD is a subclass of CVD and is unique in that the precursors and oxidants are not present in the deposition chamber at the same time. Rather, the precursor and oxidant are introduced into the chamber in a sequential and nonoverlapping way. Between the precursor and oxidant injections, a purge gas flow is maintained to remove the remaining precursor and reactant species. The combination of a precursor/oxidant pulse and purge gas flow is known as a half-cycle. Film deposition occurs through a self-saturating half-cycle since the reactions terminate by themselves once all reactive elements on the surface are consumed [9]. So, unlike CVD, where the deposition occurs on a time basis, that in ALD occurs on a cycle basis [5]. The thickness of the deposited film can be adjusted according to the number of deposition cycles. So, the self-saturating nature and cyclic deposition provide a conformal, controlled, and uniform high-quality, dense, pinhole-free thin film deposition having a thickness error of less than 1%; this film growth is independent of precursor and oxidant fluxes [7,10]. A detailed comparison of different deposition techniques including ALD is described in [9].
Location of change:
Section: Introduction
Page-1 and line 29.
Comment 5
Reviewer wrote:
CAs number, purities and provider of all the used chemicals should be added in the experimental section.
Our response:
Thanks for your valuable suggestion. We have added the information in the materials and methods section.
Corresponding change in manuscript: The above-mentioned information has added.
New:
The Chemical Abstracts Service (CAS) number for TMA is 75-24-1, while that for TEMAH is 352535-01-4. Both precursors were claimed to have a 99.9999% purity as provided by UP Chemical Co., Ltd.
Location of change:
Section: Materials and Methods
Page-2 and line 78.
Comment 6
Reviewer wrote:
The ALD equipment used to perform the deposition should be as well described.
Our response:
Thanks for your valuable suggestion. We have added the information about the ALD equipment.
Corresponding change in manuscript: The information about ALD equipment has added.
New:
The ALD system is a thermal ALD process (“Atomic Classic”, CN1), having a maximum deposition temperature range of 450°C and an allocation system of 4 sets of precursor canisters. The TMA precursor canister was kept at room temperature, and the TEMAH precursor was at 60°C. The carrier and purge gas flow rates were 300 sccm for Al2O3 deposition and 100 sccm for HfO2. The pulse times were 0.1s and 2s for Al2O3 and HfO2 precursors, respectively, while the purge time was 20s in both cases. The oxidant (H2O) pulse times were 0.1s and 0.2s for Al2O3 and HfO2 cases, respectively. All the depositions were performed at a chamber pressure of 3.6 millitorr.
Location of change:
Section: Materials and Methods
Page-2 and line 80.
Comment 7
Reviewer wrote:
Error bar should be added to all values, tables and figures given in the manuscript. Is it relevant for instance to add 4 number after the decimal point for thicknesses in table 1?
Our response:
Thanks for your valuable comment. We are sorry for the irrelevancy of thickness value representation. We have added the error bar in the figures and in the table, we have added the standard deviation data. And also, we have adjusted all the thickness value up to three decimal point.
Corresponding change in manuscript: Error bars have been added and fixed the irrelevancy between the thickness value.
Location of change:
Section: Materials and Methods; Table 1
Page-3 and line 110.
Comment 8
Reviewer wrote:
Can the authors comment on the stability of the deposited films?
Our response:
Thanks for your valuable comment. We have checked the samples with 500°C PMA conditions. All the cases, the sample behavior is ok. Also, we have checked the samples after 4 weeks of first measurement. In this case also, we have found the same result as first measurement.
Corresponding change in manuscript: No
Round 2
Reviewer 2 Report
Despite that the authors did not adress all my comments, i judge the paper acceptable now to be published in Micromachines.